# Toxicological Assessment of Oral Co-Exposure to Bisphenol A (BPA) and Bis(2-ethylhexyl) Phthalate (DEHP) in Juvenile Rats at Environmentally Relevant Dose Levels: Evaluation of the Synergic, Additive or Antagonistic Effects

**DOI:** 10.3390/ijerph18094584

**Published:** 2021-04-26

**Authors:** Roberta Tassinari, Sabrina Tait, Luca Busani, Andrea Martinelli, Mauro Valeri, Amalia Gastaldelli, Annalisa Deodati, Cinzia La Rocca, Francesca Maranghi

**Affiliations:** 1Center for Gender-Specific Medicine, Istituto Superiore di Sanità, 00161 Rome, Italy; roberta.tassinari@iss.it (R.T.); sabrina.tait@iss.it (S.T.); luca.busani@iss.it (L.B.); cinzia.larocca@iss.it (C.L.R.); 2Experimental Animal Welfare Sector, Istituto Superiore di Sanità, 00161 Rome, Italy; andrea.martinelli@iss.it (A.M.); mauro.valeri@iss.it (M.V.); 3Institute of Clinical Physiology, National Research Council, 56124 Pisa, Italy; amalia@ifc.cnr.it; 4Dipartimento Pediatrico Universitario Ospedaliero, Bambino Gesù, 00165 Rome, Italy; annalisa.deodati@opbg.net; 5Children’s Hospital, Tor Vergata University, 00133 Rome, Italy

**Keywords:** plasticizers, mixtures, in vivo, computational

## Abstract

Background: The general population (including children) is exposed to chemical mixtures. Plasticizers such as Bisphenol A (BPA) and Phthalates (mainly Bis(2-ethylhexyl) phthalate-DEHP) are widespread contaminants classified as endocrine disrupters which share some toxicological profiles and coexist in food and environment. Methods: To identify hazards of DEHP and BPA mixtures, the juvenile toxicity test—where rodents are in peripubertal phase of development, resembling childhood—was selected using exposure data from biomonitoring study in children. Biological activity and potential enhanced and/or reduced toxicological effects of mixtures due to common mechanisms were studied, considering endpoints of metabolic, endocrine and reproductive systems. The degree of synergy or antagonism was evaluated by synergy score calculation, using present data and results from the single compound individually administered. Results: In metabolic system, synergic interaction predominates in female and additive in male rats; in the reproductive and endocrine systems, the co-exposure of BPA and DEHP showed interactions mainly of antagonism type. Conclusions: The present approach allows to evaluate, for all the endpoints considered, the type of interaction between contaminants relevant for human health. Although the mode of action and biological activities of the mixtures are not completely addressed, it can be of paramount usefulness to support a more reliable risk assessment.

## 1. Introduction

Human biomonitoring studies showed that the general population (including children) is rarely exposed to a single compound, instead most frequently to chemical mixtures present in the environment (water, air, soil), in food or in consumer products. However, although progresses have been made, chemical risk assessment includes the effects of single substances, and regulatory requirements for mixtures are scarce, except for intentional mixtures such as formulated products and pharmaceuticals [1]. At present, due to lack of knowledge and experimental limitations, few data are available on the adverse effects potentially derived from long-term exposure to chemical mixtures even at low doses, around or well below regulatory reference values in humans (i.e., health-based guidance values) and animals (i.e., No Observed Adverse Effect Levels, used to derive guidance values) [2,3].

Phthalates are widespread chemicals used as softeners in polyvinyl chloride (PVC) plastics for toys, vinyl flooring and wall covering, detergents, lubricating oils, personal care products (i.e., nail polish, hair sprays, aftershave lotions, soaps, shampoos, perfumes and other). Moreover, bis(2-ethylhexyl) phthalate (DEHP) is present in food packaging and in medical equipment (injection and blood bags) [4]. Bisphenol A (BPA) is equally diffused in the environment, mainly in dental sealants, food packaging, thermal receipts, etc., since it is a polymer in polycarbonate plastics [5]. Both of these compounds are additives and not covalently bound. Therefore, they migrate and are released from plastics; for this reason, the general population can be exposed daily through ingestion, inhalation or dermal contact [6].

Both BPA and DEHP are plasticizers classified as endocrine disruptors (EDs) [7], mainly due to their weak estrogenic and anti-androgenic properties, respectively [8,9]. Furthermore, recent evidence suggests that they possess novel and sex-specific toxicological targets such as thyroid gland and metabolism linked to the window of exposure [10,11].

Considering that biomonitoring studies routinely detect both plasticizers in almost 75–90% of human samples, including children’s, toxicological studies using chemical mixtures beyond the single compound are pivotal, since they reflect the real exposure scenarios. In this respect, they can improve human risk assessment by establishing if and how chemicals might share similar modes of action or act independently on different targets. At present, few studies with binary or multi-component mixtures of BPA and phthalates have been conducted in laboratory animals. Recently, Baralić et al. [12] reported that in a mixture experiment with 50 mg/kg bw/day DEHP + 50 mg/kg bw/day dibutyl phthalate (DBP) + 25 mg/kg bw/day BPA, adult male rats presented changes in lipid profile, liver-related biochemical parameters, glucose and testosterone levels, more pronounced in the mixture. Moreover, the BPA and DEHP mixture may disturb the thyroid hormone homeostasis in female rats treated during puberty [13].

In real-life scenarios, children represent a particularly vulnerable and susceptible sub-group of the population to the effects of plasticizers, due to: (i) their action as EDs, (ii) higher exposure levels in comparison to other population groups and (iii) the critical window of development and maturation [14,15]. Scientific authorities estimated that children aged 1–6 years showed the highest DEHP daily oral intake [16,17] and that the higher BPA exposure occurs in children aged from 6 months to 10 years [18]. Moreover, Tait et al. [14,15] have recently demonstrated that, in the Italian population, DEHP and BPA urinary levels are higher in children aged 4–6 years.

Since these substances share some toxicological profiles and they usually coexist in food and environment, studies exploring the potential combined effects warrant wide attention. In this respect, in the frame of the LIFE PERSUADED project (https://lifp.iss.it/ accessed on 10 February 2021), the juvenile toxicity test was selected to identify hazards potentially derived from the exposure to binary mixtures of DEHP and BPA, using data directly derived from the LIFE PERSUADED biomonitoring study in children [10,11,14,19]. In the juvenile study, animals are treated for 28 days during the peripubertal phase of development, resembling childhood and adolescence [20,21]. Results of the studies at single exposure to BPA and DEHP have been recently published [10,11]. The present paper focuses on data obtained from oral co-exposure to binary mixtures of BPA and DEHP in male and female juvenile rats, with the aim to identify the potential enhanced and/or reduced toxicological effects due to common mechanisms, considering the metabolic, endocrine and reproductive systems as main targets. For this purpose, to evaluate and quantify the degree of synergy or antagonism between BPA and DEHP, present data and past results of single compounds [10,11] were used to calculate the synergy score of each binary mixture. Commonly, the additive effect is theoretically expected from the simple sum of two or more effects of multiple chemicals (non-interaction) and significant deviations from additive effect are classified as synergy or antagonism. According to the European Food Safety Authority (EFSA), the antagonism is defined as the toxicological or pharmacological interaction where the combined biological effect of two or more chemicals is less than expected on the basis of the simple sum of the toxicity of each substance; the synergism is the interaction in which the combined biological effect of two or more chemicals is greater than expected on the basis of the simple sum of the toxicity of each substance [22].

## 2. Materials and Methods

### 2.1. Chemicals

Bisphenol A (BPA, CAS 80-05-7; purity ≥ 99%); Bis(2-ethylhexyl) phthalate (DEHP, CAS 117-81-7; purity ≥ 99,5%) and olive oil (CAS 8001-25-0) were purchased from Sigma-Aldrich (Milano, Italy).

### 2.2. Definition of the Mixtures

As previously described in Tassinari et al. [10,11], the dose levels used in the single studies with BPA (2, 6 and 18 mg/kg body weight (bw) per day) and DEHP (9, 21 and 49 mg/kg bw per day) were defined taking into account the results of the children biomonitoring study, representing the actual exposure of Italian children and adolescents [14,15].

According to the experimental plan of the LIFE PERSUADED project, an empirical method was used to test BPA and DEHP in binary mixture. Indeed, since children of the same age display similar patterns of exposure to BPA and DEHP [14,15], a double diagonal design has been applied to combine the doses of single compounds (Figure 1), thus obtaining three mixtures with increasing concentrations of both BPA and DEHP and two mixtures combining the extreme of the two, as follows:BPA low (2 mg/kg body weight bw per day) + DEHP low (9 mg/kg bw per day) = Mix B2D9BPA middle (6 mg/kg bw per day) + DEHP middle (21 mg/kg bw per day) = Mix B6D21BPA high (18 mg/kg bw per day) + DEHP high (48 mg/kg bw per day) = Mix B18D48BPA low (2 mg/kg bw per day) + DEHP high (48 mg/kg bw per day) = Mix B2D48BPA high (18 mg/kg bw per day) + DEHP low (9 mg/kg bw per day) = Mix B18D9

A control group (C) treated with vehicle only (olive oil) was included.

By this approach, each mixture reflected both the dose level of BPA and DEHP already tested with the same experimental protocol and the exposure level directly derived from actual urinary levels in the child population. Moreover, a more reasoned and optimal number of animals has been used to meet the ‘3R’ rules and the European Union regulation on animal welfare.

### 2.3. Study Design

The animal study was managed according to the European Community Council Directive 2010/63/UE and the Italian Law 4 March 2014, n. 26 and the OECD Principles on GLP.

The design of the juvenile study has been extensively described in Tassinari et al. [10,11]. Briefly, twelve dams of Sprague–Dawley rats (Charles River—Italy) with offspring (5 pups/sex) were kept under standard laboratory conditions (22 ± 0.5 Celsius (°C) room temperature, 50–60% relative humidity, 12 h of dark-light alternation with 12–14 air changes per hour) with water and food (2018 Global Diet from Mucedola, Italy) available ad libitum. Juvenile rats (50–60 g male and 40–50 g female rats) were divided in order to obtain six groups of 10 rats/sex/group, housed in double cages and treated 5 days/week for 28 days by gavage, from postnatal day (PND) 23 (weaning) to full sexual maturity with the binary mixtures of BPA and DEHP dissolved in olive oil, one control (C) group was treated with olive oil only (vehicle). Remaining dams were not used in the experiment.

Vaginal patency and preputial separation were evaluated starting from PND32 (females) or PND40 (males). At full sexual maturity, all rats were anaesthetized with a gaseous solution of isofluorane and blood was collected by intracardiac puncture for the evaluation of serum biomarkers. After sacrifice by CO_2_ asphyxiation, necropsy and gross pathology were performed. Adrenals, liver, hypothalamic–pituitary areas and reproductive organs (testis, epididymis, uterus and ovaries) were excised and weighted. Thyroid was not weighed following excision due to the difficulty in the standardization of the tissue portion trimmed at sacrifice.

For histopathological analysis, target tissues were immediately fixed in 10% buffered formalin, except testes and epididymis fixed in Bouin’s solution.

Hypothalamic-pituitary areas (taken together) were flash frozen in liquid nitrogen and stored at −80 °C for gene expression analysis.

### 2.4. Hormone Serum Levels

Samples of whole blood were allowed to coagulate at room temperature for 1h, centrifuged twice for 15 min at 2000 rpm (Microlite Microfuge, Thermo Electron Corporation) and stored at −80 °C until use. Following hormone: 17β estradiol (E2), testosterone (T), anti-Mullerian hormone (AMH), tetraiodothyronine (T4), thyroid stimulating hormone (TSH), adiponectin and leptin were evaluated. Commercial ELISA kits were used: Rat AMH kit (CSB-E11162r), Rat adiponectin Kit (CSB-E07271r) Rat T4 Kit (CSB-E05082r) purchased from Cusabio Biotech Ltd. (Houston, TX 77054, USA) and E2 Rat kit (RTC009R), T Mouse/Rat kit (RTC001R), TSH Rat kit (RTC007R) and Leptin Mouse/Rat kit (RD291001200R) purchased from BioVendor (Brno, Czech Republic).

The manufacturer’s directions were followed, and each sample was evaluated in duplicate (absorbance 450 nm; VICTOR3 Multilabel, Perkin Elmer, Waltham, MA, USA). The unknown hormone concentrations in samples were derived using the standard curve of each hormone by the software GraphPad Prism 6.0 (GraphPad Software Inc., San Diego, CA, USA).

### 2.5. Histopathological Analysis

Fixed tissues (thyroid, adrenals, liver, uterus, ovaries, testes and epididymis) were embedded in paraffin, cut into 5 μm sections and stained with haematoxylin and eosin for light microscopy analysis (Nikon Microphot FX, Rome, Italy) with lenses from 2X to 100X. The scoring of histopathological alterations was done semi quantitatively, using a 5-point grading scale: 0: no change, grade 1: minimal, grade 2: mild, grade 3: moderate, grade 4: marked—according to distribution, severity and morphologic changes based on the criteria of Shackelford et al. [23].

Quantitative morphometrical analysis was performed using a software for image analysis (Nis-Elements D) applied to an optical microscope (Nikon Microphot FX, Rome, Italy).

Adrenal: in whole transversal section using the 2X objective, the ratio between cortex and medulla areas was calculated [24].

Thyroid: using the 10X objective, the ratio between the number of follicles and a pre-determined thyroid area (follicular density) was measured; using the 40X objective, in five randomly selected follicles/samples the following measures were performed: (a) follicle and colloid areas, (b) cell number in follicles epithelium, (c) indirect follicular cell height as mean ratio between follicle and colloid area, (d) mean ratio between follicular epithelium areas and number of nuclei; using the 64X lens, the follicular cell height (mean of five cell heights in five randomly selected follicles/sample) was measured [25].

Testis: using the 20X lens, the tubular diameters and area of the seminiferous tubules and lumen in 20 randomly-selected tubules were measured.

Uterus: using the 10X lens, a cross-section was taken from the right uterine horn, 1 cm above the uterine bifurcation, and ratio between the area of endometrium and myometrium as relative percentage of both uterine tissue components was calculated [26].

### 2.6. Gene Expression Analysis

Real-time PCR was used to evaluate gene expression of follicle stimulating hormone (FSH), luteinizing hormone (LH) and thyroid stimulating hormone (TSH) in hypothalamic–pituitary areas sex/group. All samples were mechanically disgregated by the Miccra D-1 homogenizer (ART-moderne Labortechnik, Germany) and total RNA content was extracted with the Norgen kit (Norgen, Canada) according to the manufacturer’s instructions. RNA quantity was assessed by NanoDrop (Waltham, MA, USA) and the integrity evaluated by 1% agarose gel electrophoresis. All the samples met quality criteria (integrity, A260/A280 ≥ 1.8) to proceed with real-time PCR analysis. One microgram of total RNA/each sample was reverse transcribed to cDNA using the Tetro cDNA Synthesis Kit (Quantace, UK) according to the manufacturer’s instructions. Specific primers for FSH, LH, TSH and one reference gene (glyceraldehyde 3-phosphate dehydrogenase (GAPDH)) (Appendix A), were designed using the Primer-BLAST web application (www.ncbi.nlm.nih.gov/tools/primer-blast, accessed on 9 April 2021) and purchased from Invitrogen (Life Technologies, London, UK). The SensiMix Plus SYBR Kit (Quantace) was used to perform real-time PCR assays running experiments in duplicate on a Bioer LineGene 9600 (Bioer, Hangzhou, China) with the following thermal program: one cycle at 94 °C for 10 min; 40 cycles at 94 °C for 10 s, 58 °C for 10 s and 72 °C for 10 s; one melting cycle from 55 to 94 °C to verify amplification products. Threshold cycles (Ct) were calculated by the LineGene 9620 software (Bioer, Hangzhou, China) expressing data as ΔΔCt ± SEM values for each target gene with control samples as calibrator and GAPDH as reference gene.

### 2.7. Statistical Analysis

Stata ver.14.2 (StataCorp, Lakeway Drive College Station, TX, USA) and GraphPah PRISM 6 (GraphPad Software Inc., La Jolla, CA, USA) were used for the statistical analysis. Non-parametric Kruskal-Wallis statistics was applied to all the continuous variables to assess statistical differences among groups, followed by the post-hoc Mann-Whitney’s pair-wise test comparison, where appropriate. For categorical variables (histological endpoints), data were expressed as proportions of quantal data; pair-wise comparisons of treated groups vs C group were performed by the two-tailed Fisher exact test. The Mantel–Haenszel χ^2^ trend test was used to identify dose–response trends. Significance level was set at *p* < 0.05.

### 2.8. Synergy Score Calculation

Data of both male and female rats for each endpoint—in treated groups as well as results from previous experiments with BPA and DEHP individually administered [10,11]—were normalized to the corresponding C group values.

In order to obtain a complete matrix to calculate synergy scores, the missing responses were firstly estimated by applying the nonnegative matrix factorization (cNMF) algorithm implemented in the SynergyFinder web application [27] (https://synergyfinder.fimm.fi, accessed on 29 March 2021).

Synergy scores were then calculated with Combenefit [28] using both the Loewe [29] and the Bliss synergy models [30], since the Loewe is more appropriate when drugs affect the same pathways thus displaying additivity, whereas the Bliss is more appropriate when compounds act independently affecting different pathways. Since BPA and DEHP may have either common or different targets [10,11], we decided to perform both the analyses. According to SynergyFinder, the results were interpreted as follows: values in the range from –10 to 10, the interaction was considered additive; less than –10, the interaction was considered antagonistic, and less than –25, strong antagonistic; more than 10 the interaction was considered synergic, and more than 25 strong synergic.

## 3. Results

No death or adverse effects on health have been recorded in both male and female rats during the treatment period.

### 3.1. Metabolic System

The main effects on metabolic system are summarized in Table 1. The calculated synergy scores of the binary mixtures obtained with the Bliss and the Loewe models were mostly overlapped, therefore only the results of Loewe model are shown in Figure 2.

The bw gain was unaffected by the treatment of BPA and DEHP binary mixtures in female rats; in male rats, only Mix B6D21 induced a significant increase in comparison to C group, with additive interaction. Feed consumption was significantly decreased in female rats by Mix B2D9 and Mix B18D9 with synergic and additive effects, respectively, whereas it was unaffected in male rats.

Liver absolute weight was significantly increased in male rats in Mix B2D9, Mix B6D21 and Mix B18D48, all displaying additive effects, while it was unaffected in female rats in comparison to C group. Liver relative weight was significantly increased in male rats only by Mix B18D48, with an additive score as for the absolute weight; otherwise, a significant decrease of liver relative weight was observed in female rats treated with Mix B18D9 displaying an additive effect.

Liver histopathological analysis showed a significant increase in the extramedullary hematopoiesis following Mix B6D21 treatment in comparison to C group in both sexes with synergic score. In liver of females, significantly increased inflammatory cell foci were present only in the Mix B2D9 treated group, associated to a very strong antagonist effect. A significantly reduced hepatocytic vacuolation was induced by Mix B18D9 and Mix B18D48 in male rats, associated to a strong synergism.

Adiponectin serum levels were significantly increased in Mix B2D48 with a strong antagonist effect in both sexes, and significantly decreased in Mix B18D48, associated to synergism, in female rats. A different pattern was observed in leptin serum levels which were significantly decreased by Mix B18D48 and Mix B2D48 in both sexes; Mix B2D48 displayed a strong synergism in both sexes whereas Mix B18D48 had a strong antagonist score in males and an additive score in females. In addition, Mix B18D9 significantly decreased leptin levels with synergic effect in female rats only.

### 3.2. Endocrine System

The main endocrine effects are summarized in Table 2. The calculated synergy scores of the binary mixtures obtained with the Bliss and the Loewe models were mostly overlapped, therefore only the results of Loewe model are shown in Figure 3.

In female rats, adrenal absolute and relative weight was significantly increased by Mix B2D48 and Mix B18D48 in comparison to C group associated to an antagonistic interaction. The same antagonism was observed also in males for the same mixtures but without any significant difference on adrenal absolute and relative weight in comparison to C.

Mix B18D48 significantly increased both cortex and medulla areas in female rats only, associated to a strong antagonism. No treatment-related alterations were present in adrenals of male rats.

Histomorphometrical analysis of thyroid showed a significantly increased ratio of follicular epithelium areas in Mix B2D48 and Mix B18D48 of female treated groups with an antagonistic interaction; these same binary mixtures significantly increased in female rats the ratio between follicular epithelium areas and the number of cells with an additive effect. On the contrary, in male rats, a significant decrease of the thyroid follicular epithelium area in Mix B18D48 group in comparison to C group, associated to a synergic interaction, was observed.

No treatment-related alterations were present in TSH and T4 serum levels and in TSH gene expression in both male and female rats in comparison to C group (data not shown).

### 3.3. Reproductive Systems

The main reproductive effects are summarized in Table 3. The calculated synergy scores of the binary mixtures obtained with the Bliss and the Loewe models were mostly overlapped, therefore only the results of Loewe model are shown in Figure 4.

Timing of vaginal opening was significantly delayed by Mix B18D48 with an additive effect, whereas preputial separation was not affected by any treatment.

Uterus absolute weight was unaffected whereas its relative weight was significantly increased by Mix B18D9 associated to a synergic effect.

Absolute and relative weight of ovaries, testes and epididymis was unaffected.

In male rats, E2 serum levels were significantly increased by Mix B2D9 and Mix B2D48 displaying a strong antagonism, as for all the other tested binary mixtures; on the contrary, in female rats, E2 serum levels were significantly decreased by Mix B18D9 with a strong synergism, while all the other mixtures showed antagonism as in males but to a lower extent.

T serum levels were significantly increased in female rats by Mix B2D48 associated to a strong antagonistic interaction, present also in all the other mixtures. In male rats, T levels were unaffected although displaying similar antagonism of the mixtures.

AMH serum levels were significantly increased in male rats by Mix B6D21 and Mix B18D9 associated with a strong antagonism, observable also for the other mixtures; similarly, in female rats, Mix B2D48 and Mix B6D21 increased AMH serum levels displaying strong antagonism.

Testis histopathological analysis showed a significantly increased germ cell degeneration induced by Mix B6D21 associated to a strong antagonism. Testis transversal diameter was significantly increased by Mix B2D9 and Mix B18D9 displaying antagonistic interaction.

No treatment-related histopathological alterations were present in epididymis, ovary and uterus.

In female rats, a significant down-regulation of FSH gene expression was induced by Mix B18D48 associated with a strong synergism. No effect was observed in male rats.

A significant up-regulation of LH gene expression was induced in male rats by Mix B2D9 and Mix B2D48 and, in female, rats by Mix B2D9, Mix B2D48 and Mix B18D9, associated to a strong antagonism.

## 4. Discussion

In the hazard identification of mixtures, the use of adequate model systems (in vivo, in vitro, in silico and computational) is critical for better understanding the toxicity of chemical mixtures [31]. In such context, the use of computational tools as SynergyFinder and Combenefit may be applied to investigate the degree of a combination effect, easily visible as a synergy landscape map over the dose matrix. So far, these tools were used in in vitro screening studies, to highlight novel synergistic drug combinations in pre-clinical models (e.g., cell lines or primary cells derived from patients), and to better understand the mechanisms of efficacy or resistance of combined treatments [27,32]. In the present paper, for the first time, computational tools were applied to analyze the combined effect of mixtures of chemicals commonly present in food and environment; in addition, the study shows other aspects of novelty: (a) the relevance of results for risk assessment in children since the dose levels of BPA and DEHP have been selected starting from the real exposure scenario in Italian population; (b) the evaluation of chemical mixtures in an in vivo model, according to sex, and (c) data management using cutting-edge tools. Since no other study, up to this point, has applied such an approach to evaluate the complex interactions of contaminants in mixtures and the toxicological consequences of the exposure, the present data are discussed with results obtained by the same compounds—individually administered—derived from the studies of Tassinari et al. [10,11]. Indeed, BPA, DEHP and their mixtures belong to the same experimental plan, and they were tested all together; the Authors decided to discuss the data separately to better reveal the relevance of the results.

Concerning the metabolic system, different pathways of interaction prevailed according to sex. In fact, in female rats the majority of mixtures exerting significant effects, in comparison to the concurrent C, displayed synergic interaction, followed by additive and antagonistic interactions; in male rats, the majority of mixtures displayed additive interaction followed by synergic and antagonistic (Table 4).

In the general toxicity parameters in both sexes, the results of single compounds [10,11] clearly identified the additive—even potentiated—effect recorded for all the mixtures. Concerning the histopathological effects, liver has shown to be target of both compounds individually administered, but all single BPA dose levels and only the highest DEHP dose levels increased inflammatory cell foci in female rats [10,11]; the co-exposure led to a significant increase only in Mix B2D9, confirming the strong antagonism. Moreover, the extramedullary hematopoiesis was only induced by strong synergic effects as in Mix B6D21; the individual administration of both compounds didn’t cause the same alteration, thus supporting the synergism. Interestingly, for this endpoint, Mix B6D21 was the only mixture displaying a synergism since all the other mixtures displayed antagonism. Hepatocytic vacuolization in male rats is apparently driven by DEHP since, in individual administration, it exerted the same effect at all doses tested, significantly at the higher levels [11]. BPA alone did not [10], therefore the synergism between compounds might explain the effect. Overall, metabolic data indicate that the two plasticizers, triggering different pathways, may determine overlapping or compensatory derangements. Indeed, both compounds are known to induce liver oxidative stress, but BPA seems to mainly induce non-alcoholic fatty liver disease [33,34] whereas DEHP, although similarly involved in lipid metabolism disorders, is a potential carcinogen through the activation of the peroxisome proliferating receptor gamma [35]. Concerning serum metabolic biomarkers, none of the mixtures tested apparently induced metabolic effects with similar pattern as recorded with DEHP individually administered [11] in both sexes, or at the highest dose of BPA, predominantly in males [10]. Indeed, only Mix B2D48 in both sexes seemed to induce obesogenic effects due to the increased levels of the proinflammatory leptin and decreased levels of anti-inflammatory adiponectine [36]. Nevertheless, no other effects supporting this mechanism are present with same mixture. Scarce data are available on toxicity of plasticizer mixtures in metabolic system; a recent study compared the subacute toxic effects of low doses of DEHP, DBP (50 mg/kg bw/day) and BPA (25 mg/kg bw/day) administered alone with the effects of their mixture analyzing different biochemical, hormonal, and hematological parameters in male rats. The mixture induced significant changes in lipid profile, liver-related biochemical parameters and glucose level [12]. In the present study, the use of computational tools could help in identifying the specific relationship among the compounds present in the mixture in order to better characterize the effects recorded on different endpoints.

The endocrine system is the target of co-exposure of BPA and DEHP when one or both compounds are present at the higher dose levels with a clear sex-specificity; in fact, in male rats, the significant effect occurred in mixtures showed synergic interaction, whereas in female rats—which appeared to be more susceptible than male rats—the significant effects showed antagonistic or additive interactions (Table 4). The adrenals were targeted only in female rats and the effect is apparently driven by BPA, which induced the same effect when administered individually at the same dose level [10]. Interestingly, the co-exposure generated an antagonistic score which apparently didn’t influence the BPA effect.

The only significant endocrine effect recorded in male rats was exerted in thyroid when both compounds were at the highest dose tested with synergic interaction. Thyroid was the target of DEHP at the highest dose level, but no effects were present with BPA alone [10,11]. The combined effects of DEHP and BPA on thyroid function during puberty was also explored in female Sprague-Dawley rats treated per os by gavage from postnatal day 28 to 70—similar to the present juvenile experiment—with DEHP (0, 150 and 750 mg/kg/day) and BPA (0, 20 and 100 mg/kg/day), alone or in mixture. The study concluded that DEHP and BPA in mixture may disturb the thyroid hormone homeostasis and consequently the development of the thyroid during puberty [13].

In female reproductive system, the majority of mixtures with significant effects showed antagonistic interaction, followed, to a lesser extent, by additivity and synergism (Table 4). Indeed, it is noteworthy that the additive co-exposure of BPA and DEHP at the highest dose delayed vaginal opening since such alteration was not present when both compounds were administered individually [10,11]. The same mixture didn’t exert any other effect in reproductive function. Interestingly, the significant effects induced by mixtures in female reproductive system didn’t find any correspondence with those induced by BPA and DEHP individually administered [10,11]. Scarce data are available on reproductive effects of BPA and DEHP co-exposure in female rats. Dams exposed to an environmentally relevant phthalate mixture (35% diethyl phthalate, 21% DEHP, 15% DBP, 15% di-isononyl phthalate, 8% di-isobutyl phthalate and 5% benzylbutyl phthalate) showed increased uterus weight in female offspring up to the F3 generation [37]. However, this study is of limited relevance for the comparison with the present data since the mixture does not contain BPA and the window of exposure is different. Overall, the present study is the first to point out toxicity on the female reproductive system, in particular on uterus weight and on hormone serum levels, due to oral co-exposure to BPA and DEHP in rats exposed during juvenile life-stage.

In the male reproductive system, BPA and DEHP administered in binary mixtures always showed antagonistic scores for all the significantly altered endpoints (Table 4), and this feature can account for the lack of correlation between effects due to exposure to single compounds and the mixtures. Male rats treated per os with DEHP (50 mg/kg bw/day), DBP (50 mg/kg bw/day), BPA (25 mg/kg bw/day) and their MIX (50 mg/kg bw/day DEHP + 50 mg/kg bw/day DBP + 25 mg/kg bw/day BPA) showed more evident testicular toxicity in the MIX group (desquamated germinal epithelium cells, enlarged cells with hyperchromatic nuclei, multinucleated cells and intracytoplasmic vacuoles) in comparison with the individual chemicals, while effects on redox status were either more prominent, or present only in the MIX group, confirming that the male reproductive system might be more susceptible following exposure to chemicals in mixtures than individually [38]. Moreover, when orally administered during pregnancy and lactation, the mixture of BPA (50 mg/kg/day), DEHP (30 mg/kg/day) and their binary mixture (50 mg/kg/day BPA + 30 mg/kg/day DEHP) shows more dramatic changes in both testicular structure and cell death [39].

BPA and DEHP are classified as EDs that mimic, block or interfere with the endocrine system, inducing a diverse array of health issues. BPA is a weak estrogen-like compound able to link to estrogen receptor alpha [40], whereas DEHP can disrupt androgen receptor signaling [41]. Results obtained with the computational tools contributed to put into evidence that the interactions between compounds are mainly of antagonism type, and it may be in part explained with the different mechanisms at receptor level. On the other hand, the complexity of pathways involved in hormone homeostasis needs more mechanistic studies to characterize the toxicological effects of mixtures.

## 5. Conclusions

The assessment of chemical mixtures is a complex topic for toxicologists, regulators, and the public. The interpretation of data derived from toxicological studies using environmental contaminants in mixtures—although of great relevance for risk assessment—is still affected by several grades of uncertainty. The present study, based on the use of computational methods on in vivo data, allowed to highlight the type of interaction between compounds for any single endpoint considered; this information can be of paramount usefulness in the evaluation of the effects. On the other hand, the approach didn’t provide specific support for the identification of mode of action and biological activities of the mixtures, and both synergic and antagonistic effects were recorded according to the different functional systems. However, this provides further support to the hypothesis that different underlying modes of action are exerted by the two compounds, that sometimes converge on the same endpoints. Overall, the use of computational tools for the identification of the interactions between mixtures can be considered useful also in in vivo studies for a more reliable toxicological risk assessment of mixtures.

## Figures and Tables

**Figure 1 ijerph-18-04584-f001:**
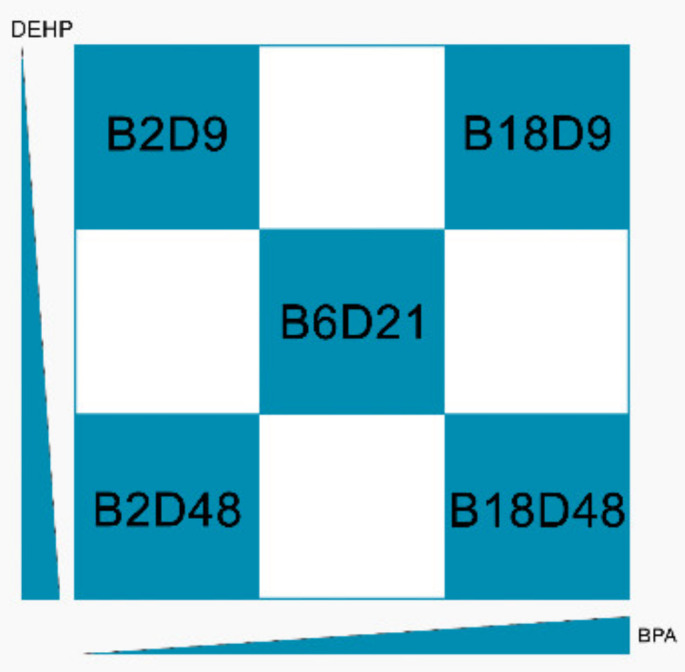
Combination design of Bisphenol A (BPA) and Bis(2-ethylhexyl) phthalate (DEHP) dose levels.

**Figure 2 ijerph-18-04584-f002:**
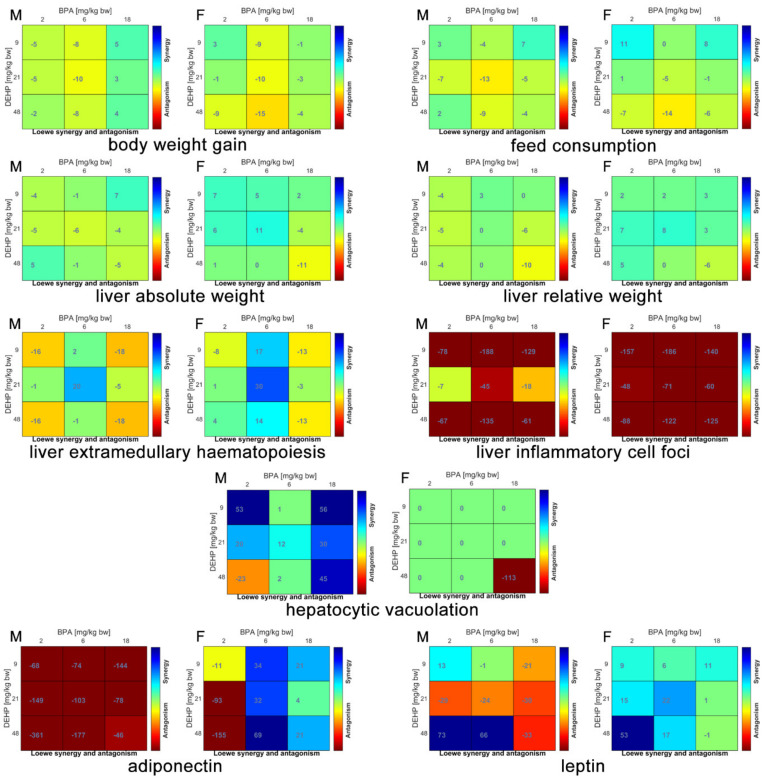
Loewe model synergy score matrices of metabolic effects in male and female rats orally treated for 28 days with binary mixtures of BPA and DEHP:Mix B2D9 (BPA 2 mg/kg bw per day + DEHP 9 mg/kg bw per day), Mix B6D21 (BPA 6 mg/kg bw per day + DEHP 21 mg/kg bw per day), Mix B18D48 (BPA 18 mg/kg bw per day + DEHP 48 mg/kg bw per day), Mix B2D48 (BPA 2 mg/kg bw per day + DEHP 48 mg/kg bw per day) and Mix B18D9 (BPA 18 mg/kg bw per day + DEHP 9 mg/kg bw per day). M: Male; F: Female.

**Figure 3 ijerph-18-04584-f003:**
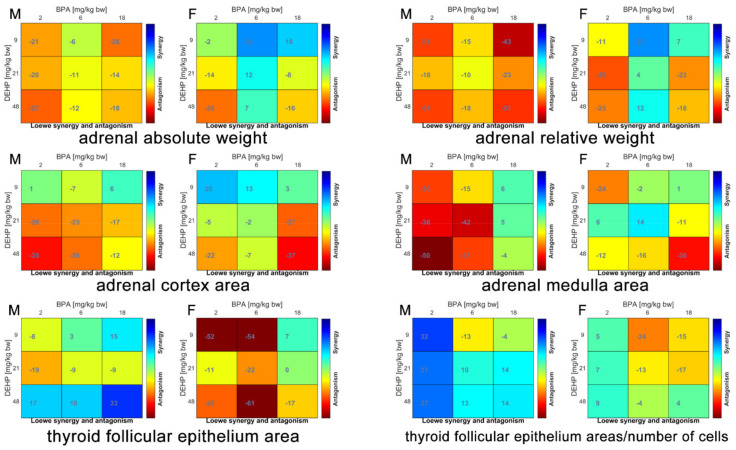
Loewe model synergy score matrices of endocrine effects in male and female rats orally treated for 28 days with binary mixtures of BPA and DEHP: Mix B2D9 (BPA 2 mg/kg bw per day + DEHP 9 mg/kg bw per day), Mix B6D21 (BPA 6 mg/kg bw per day + DEHP 21 mg/kg bw per day), Mix B18D48 (BPA 18 mg/kg bw per day + DEHP 48 mg/kg bw per day), Mix B2D48 (BPA 2 mg/kg bw per day + DEHP 48 mg/kg bw per day) and Mix B18D9 (BPA 18 mg/kg bw per day + DEHP 9 mg/kg bw per day). M: Male; F: Female.

**Figure 4 ijerph-18-04584-f004:**
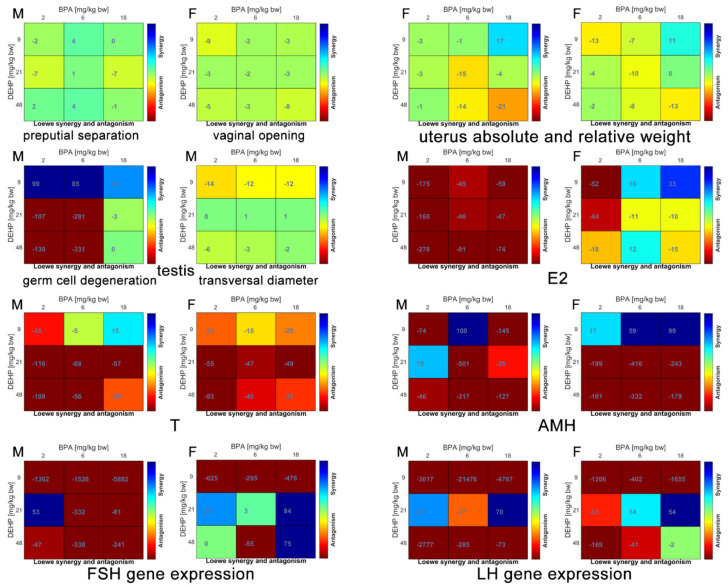
Loewe model synergy score matrices of reproductive effects in male and female rats orally treated for 28 days with binary mixtures of BPA and DEHP: Mix B2D9 (BPA:2 mg/kg bw per day + DEHP 9 mg/kg bw per day), Mix B6D21 (BPA 6 mg/kg bw per day + DEHP 21 mg/kg bw per day), Mix B18D48 (BPA 18 mg/kg bw per day + DEHP 48 mg/kg bw per day), Mix B2D48 (BPA 2 mg/kg bw per day + DEHP 48 mg/kg bw per day) and Mix B18D9 (BPA 18 mg/kg bw per day + DEHP 9 mg/kg bw per day). M: Male; F: Female.

**Table 1 ijerph-18-04584-t001:** Metabolic effects in male and female rats orally treated for 28 days with binary mixtures of BPA and DEHP: C (0 mg/kg bw per day–olive oil); Mix B2D9 (BPA 2 mg/kg bw per day + DEHP 9 mg/kg bw per day), Mix B6D21 (BPA 6 mg/kg bw per day + DEHP 21 mg/kg bw per day), Mix B18D48 (BPA 18 mg/kg bw per day + DEHP 48 mg/kg bw per day), Mix B2D48 (BPA 2 mg/kg bw per day + DEHP 48 mg/kg bw per day) and Mix B18D9 (BPA 18 mg/kg bw per day + DEHP 9 mg/kg bw per day). Kruskal-Wallis ** *p* < 0.01; * *p* < 0.05. § two-tailed Fisher exact test *p* < 0.05. PND: postnatal day. M: mean. SEM: standard error of the mean.

Endpoint	Sex	C	B2D9	B6D21	B18D48	B2D48	B18D9
Body weight gain (g; M ± SEM)	Male	296 ± 5 (*n* = 9)	311 ± 16 (*n* = 8)	331 ± 10 * (*n* = 8)	300 ± 11 *n* = 8)	304 ± 9 (*n* = 7)	292 ± 10 (*n* = 8)
	Female	162 ± 4 (*n* = 9)	165 ± 10 (*n* = 7)	176 ± 4 (*n* = 7)	167 ± 10 (*n* = 8)	176 ± 6 (*n* = 8)	157 ± 6 (*n* = 8)
Feed consumption (g; M ± SEM)	Male	19.66 ± 0.38 (*n* = 9)	19.25 ± 0.49 (*n* = 8)	21.60 ± 1.14 (*n* = 8)	20.42 ± 0.84 (*n* = 8)	19.51 ± 0.8 (*n* = 7)	18.47 ± 0.82 (*n* = 8)
	Female	14.85 ± 0.85 (*n* = 9)	13.03 ± 0.26 * (*n* = 7)	15.52 ± 0.32 (*n* = 7)	14.66 ± 0.52 (*n* = 8)	16.511.18 (*n* = 8)	12.57 ± 0.78 * (*n* = 8)
Liver absolute weight (g; M ± SEM)	Male	10.72 ± 1.60 (*n* = 9)	15.36 ± 1.16 *(*n* = 8)	15.98 ± 0.55 **(*n* = 8)	16.81 ± 0.81 **(*n* = 7)	13.82 ± 0.75 (*n* = 6)	15.28 ± 0.88 (*n* = 7)
	Female	7.20 ± 0.83 (*n* = 9)	8.49 ± 0.59 (*n* = 7)	8.22± 0.26 (*n* = 7)	7.40 ± 0.27 (*n* = 8)	8.88 ± 0.26 (*n* = 8)	8.88 ± 0.68 (*n* = 8)
Liver relative weight (×100; M ± SEM)	Male	4.05 ± 0.15 (*n* = 9)	4.23 ± 0.36 (*n* = 8)	4.41 ± 0.08 (*n* = 8)	4.63 ± 0.13 * (*n* = 7)	3.69 ± 0.54 (*n* = 6)	4.06 ± 0.21 (*n* = 7)
	Female	4.00 ± 0.12 (*n* = 9)	3.78 ± 0.15 (*n* = 7)	3.62 ± 0.12 (*n* = 7)	3.95 ± 0.13 (*n* = 8)	3.84 ± 0.10 (*n* = 8)	3.53 ± 0.08 * (*n* = 8)
Liver extramedullary hematopoiesis	Male	0/9	0/8	3/8 §	0/7	0/8	0/8
	Female	0/9	0/7	3/7 §	0/5	1/8	0/8
Liver inflammatory cell foci	Male	5/9	7/8	5/8	7/7 §	6/8	5/8
	Female	3/9	7/7 §	4/7	4/5	5/8	6/8
Liver hepatocytic vacuolation	Male	5/9	1/8	2/8	0/7 §	3/8	0/8 §
	Female	1/9	0/7	0/7	0/5	0/8	1/8
Adiponectin serum levels (ng/mL)	Male	474 ± 116 (*n* = 6)	531 ± 121 (*n* = 5)	772 ± 130 (*n* = 5)	485 ± 85 (*n* = 5)	1714 ± 144 ** (*n* = 7)	667 ± 168 (*n* = 5)
	Female	838 ± 173 (*n* = 5)	499 ± 166 (*n* = 6)	367 ± 92 (*n* = 5)	2127 ± 184 ** (*n* = 8)	1004 ± 390 * (*n* = 6)	838 ± 173 (*n* = 5)
Leptin serum levels (ng/mL)	Male	7.65 ± 0.62 (*n* = 5)	6.12 ± 1.27 (*n* = 6)	6.45 ± 1.12 (*n* = 6)	6.50 ± 1.34 * (*n* = 6)	0.15 ± 0.02 * (*n* = 7)	5.03 ± 0.64 (*n* = 6)
	Female	4.44 ± 0.84 (*n* = 5)	3.24 ± 0.53 (*n* = 6)	1.90 ± 0.60 (*n* = 6)	1.47 ±0.44 ** (*n* = 6)	0.14 ± 0.02 ** (*n* = 7)	1.86 ± 0.46 * (*n* = 6)

**Table 2 ijerph-18-04584-t002:** Endocrine effects of male and female rats orally treated for 28 days with binary mixtures of BPA and DEHP: C (0 mg/kg bw per day–olive oil); Mix B2D9 (BPA 2 mg/kg bw per day + DEHP 9 mg/kg bw per day), Mix B6D21 (BPA 6 mg/kg bw per day + DEHP 21 mg/kg bw per day), Mix B18D48 (BPA 18 mg/kg bw per day + DEHP 48 mg/kg bw per day), Mix B2D48 (BPA 2 mg/kg bw per day + DEHP 48 mg/kg bw per day) and Mix B18D9 (BPA 18 mg/kg bw per day + DEHP 9 mg/kg bw per day). Kruskal-Wallis ** *p* < 0.01; * *p* < 0.05. PND: postnatal day. M: mean. SEM: standard error of the mean.

Endpoint	Sex	C	B2D9	B6D21	B18D48	B2D48	B18D9
Adrenal absolute weight (g; M ± SEM)	Male	0.07 ± 0.02 (*n* = 9)	0.12 ± 0.02 (*n* = 8)	0.12 ± 0.07 (*n* = 8)	0.11 ± 0.01 (*n* = 8)	0.11 ± 0.03 (*n* = 7)	0.12 ± 0.01 (*n* = 8)
	Female	0.07 ± 0.01 (*n* = 9)	0.09 ± 0.04 (*n* = 7)	0.09 ± 0.02 (*n* = 6)	0.10 ± 0.01* (*n* = 8)	0.12 ± 0.03 ** (*n* = 8)	0.09 ± 0.02 (*n* = 8)
Adrenal relative weight (X100; M ± SEM)	Male	0.03 ± 0.01 (*n* = 9)	0.03 ± 0.01 (*n* = 8)	0.03 ± 0.02 (*n* = 8)	0.03 ± 0.01 (*n* = 8)	0.03 ± 0.01 (*n* = 7)	0.04 ± 0.01 (*n* = 7)
	Female	0.04 ± 0.00 (*n* = 9)	0.04 ± 0.01 (*n* = 7)	0.04 ± 0.01 (*n* = 6)	0.05 ± 0.01 (*n* = 8)	0.05 ± 0.01 * (*n* = 8)	0.04 ± 0.01 (*n* = 8)
Adrenal cortex area (mm^2^; M ± SEM)	Male	4.07 ± 0.97 (*n* = 5)	4.60± 0.29 (*n* = 8)	5.35 ± 0.43 (*n* = 8)	4.98 ± 0.29 (*n* = 7)	5.50 ± 0.58 (*n* = 6)	4.14 ± 0.17 (*n* = 6)
	Female	3.75 ± 1.08 (*n* = 5)	4.21± 0.44 (*n* = 7)	4.51 ± 0.46 (*n* = 6)	6.28 ± 0.29 * (*n* = 6)	5.94 ± 0.57 (*n* = 6)	4.93 ± 0.67 (*n* = 6)
Adrenal medulla area (mm^2^; M ± SEM)	Male	0.53 ± 0.14 (*n* = 5)	1.02 ± 0.18 (*n* = 8)	0.88 ± 0.17 (*n* = 8)	0.61 ± 0.07 (*n* = 7)	0.73 ± 0.12 (*n* = 6)	0.67 ± 0.15 (*n* = 6)
	Female	0.59 ± 0.16 (*n* = 5)	0.84 ± 0.12 (*n* = 7)	0.54 ± 0.10 (*n* = 6)	0.91 ± 0.04 * (*n* = 6)	0.81 ± 0.21 (*n* = 6)	0.58 ± 0.08 (*n* = 6)
Thyroid follicular epithelium areas (mm^2^; M ± SEM)	Male	435 ± 32 (*n* = 7)	397 ± 100 (*n* = 6)	504 ± 28 (*n* = 7)	305 ± 17 ** (*n* = 8)	415 ± 62 (*n* = 5)	355 ± 101 (*n* = 6)
	Female	326 ± 19 (*n* = 7)	746 ± 237 (*n* = 6)	660 ± 235 (*n* = 7)	1854 ± 536 (*n* = 8)	470± 103 (*n* = 5)	285 ± 65 (*n* = 5)
Thyroid follicular epithelium areas/number of cells (mm^2^/*n*; M ± SEM)	Male	19.10 ± 1.91 (*n* = 7)	34.23 ± 11.86 (*n* = 6)	22.06 ± 1.46 (*n* = 7)	18.16 ± 1.26 (*n* = 8)	21.41 ± 1.58 (*n* = 5)	21.08 ± 4.61 (*n* = 6)
	Female	17.42 ± 1.51 (*n* = 7)	38.36 ± 12.10 (*n* = 6)	26.25 ± 6.97 (*n* = 7)	47.04 ± 17.44 * (*n* = 8)	24.43 ± 2.44 * (*n* = 5)	11.53 ± 2.20 (*n* = 5)

**Table 3 ijerph-18-04584-t003:** Reproductive effects of male and female rats orally treated for 28 days with binary mixtures of BPA and DEHP: C (0 mg/kg bw per day–olive oil); Mix B2D9 (BPA 2 mg/kg bw per day + DEHP 9 mg/kg bw per day), Mix B6D21 (BPA 6 mg/kg bw per day + DEHP 21 mg/kg bw per day), Mix B18D48 (BPA 18 mg/kg bw per day + DEHP 48 mg/kg bw per day), Mix B2D48 (BPA 2 mg/kg bw per day + DEHP 48 mg/kg bw per day) and Mix B18D9 (BPA 18 mg/kg bw per day + DEHP 9 mg/kg bw per day). Kruskal-Wallis ** *p* < 0.01; * *p* < 0.05. § two-tailed Fisher exact test *p* < 0.05. PND: postnatal day. M: mean. SEM: standard error of the mean.

Endpoint	Sex	C	B2D9	B6D21	B18D48	B2D48	B18D9
Timing of preputial separation/vaginal opening (day; M ± SEM)	Male	41.11 ± 0.48 (*n* = 9)	43.13 ± 0.99 (*n* = 8)	41.13 ± 0.61 (*n* = 8)	41.25 ± 0.31 (*n* = 8)	40.50 ± 0.38 (*n* = 8)	41.43 ± 0.57 (*n* = 7)
	Female	34.78 ± 0.52 (*n* = 9)	36.29 ± 0.89 (*n* = 7)	34.86 ± 0.77 (*n* = 7)	36.75 ± 0.77 * (*n* = 8)	35.88 ± 0.58 (*n* = 8)	34.88 ± 0.30 (*n* = 8)
Uterus absolute weight (g; M ± SEM)	Female	0.58 ± 0.14 (*n* = 9)	0.70 ± 0.24 (*n* = 7)	0.63 ± 0.12 (*n* = 7)	0.64 ± 0.26 (*n* = 8)	0.46 ± 0.15 (*n* = 8)	0.58 ± 0.14 (*n* = 8)
Uterus relative weight (X100; M ± SEM)	Female	0.27 ± 0.06 (*n* = 9)	0.30 ± 0.10 (*n* = 7)	0.29 ± 0.08 (*n* = 7)	0.28 ± 0.10 (*n* = 8)	0.22 ± 0.07 (*n* = 8)	0.27 ± 0.06 * (*n* = 8)
Estradiol (pg/mL; M ± SEM)	Male	2.69 ± 0.43 (*n* = 6)	5.83 ± 0.46 ** (*n* = 5)	2.26 ± 0.29 (*n* = 5)	3.51 ± 0.98 (*n* = 5)	8.47 ± 1.08 ** (*n* = 6)	3.40 ± 0.58 (*n* = 5)
	Female	20.49 ± 3.74 (*n* = 7)	28.50 ± 8.05 (*n* = 5)	16.96 ± 5.79 (*n* = 5)	11.38 ± 3.79 (*n* = 5)	14.16 ± 3.05 (*n* = 6)	3.75 ± 1.01 * (*n* = 5)
Testosterone (ng/mL; M ± SEM)	Male	2.50 ± 0.59 (*n* = 6)	3.09 ± 0.82 (*n* = 6)	4.61 ± 1.14 (*n* = 6)	3.14 ± 0.94 (*n* = 6)	4.04 ± 0.48 (*n* = 7)	1.84 ± 0.45 (*n* = 6)
	Female	0.92 ± 0.11 (*n* = 6)	0.89 ± 0.19 (*n* = 6)	1.25 ± 0.17 (*n* = 6)	0.96 ± 0.12 (*n* = 6)	1.51 ± 0.09 ** (*n* = 7)	0.85 ± 0.11 (*n* = 6)
Anti-Mullerian Hormone (ng/mL; M ± SEM)	Male	1.92 ± 0.42 (*n* = 5)	3.18 ± 0.51 (*n* = 5)	7.63 ± 1.74 * (*n* = 5)	3.94 ± 0.84 (*n* = 5)	2.26 ± 0.21 (*n* = 7)	3.90 ± 0.28 * (*n* = 7)
	Female	8.61 ± 1.25 (*n* = 6)	9.04 ± 3.92 (*n* = 5)	39.71 ± 11.69 ** (*n* = 6)	21.45 ± 8.23 (*n* = 5)	24.95 ± 4.73 ** (*n* = 7)	23.53 ± 20.39 (*n* = 5)
Testis germ cell degeneration	Male	1/9	0/6	6/8§	1/6	2/7	0/7
Testis transversal diameter (mμ; M ± SEM)	Male	165.21 ± 4.65 (*n* = 7)	189.86 ± 4.45 ** (*n* = 6)	162.67 ± 3.41 (*n* = 8)	164.44 ± 10.25 (*n* = 8)	172.39 ± 2.57 (*n* = 7)	186.88 ± 6.73 * (*n* = 6)
Follicle- stimulating hormone gene expression (ΔΔct; M ± SEM)	Male	0 ± 0.30 (*n* = 5)	3.38 ± 1.64 (*n* = 5)	1.81 ± 0.58 (*n* = 5)	1.50 ± 0.45 (*n* = 5)	−0.35 ± 0.79 (*n* = 5)	5.95 ± 0.34 (*n* = 5)
	Female	0 ± 0.30 (*n* = 5)	2.62 ± 1.01 (*n* = 5)	−0.21 ± 0.44 (*n* = 5)	−3.33 ± 0.97 * (*n* = 5)	0.09 ± 0.18 (*n* = 5)	0.38 ± 2.81 (*n* = 5)
Luteinizing hormone gene expression (ΔΔct; M ± SEM)	Male	0 ± 0.15 (*n* = 5)	5.24 ± 0.57 * (*n* = 5)	0.26 ± 0.20 (*n* = 5)	0.64 ± 0.31 (*n* = 5)	4.88 ± 0.54 * (*n* = 5)	5.03 ± 0.72 (*n* = 5)
	Female	0 ± 0.28 (*n* = 5)	4.26 ± 0.61 * (*n* = 5)	−0.45 ± 0.27 (*n* = 5)	−0.36 ± 0.21 (*n* = 5)	1.23 ± 0.33 * (*n* = 5)	4.04 ± 0.88 * (*n* = 5)

**Table 4 ijerph-18-04584-t004:** Percentage of synergistic, additive or antagonistic interactions in metabolic, endocrine and reproductive systems of male and female rats orally treated for 28 days with binary mixtures of BPA and DEHP.

Endpoint	Sex	Synergism	Additive	Antagonism
Metabolic system	Male	33%	42%	25%
	Female	50%	30%	20%
Endocrine system	Male	100%		
	Female		43%	57%
Reproductive system	Male			100%
	Female	30%	10%	60%

## Data Availability

Data available on request due to restrictions. The data presented in this study are available on request from the corresponding author. The data are not publicly available due to the data policy of PERSUADED project. The data presented in this study are available on request from the corresponding author. The data are not publicly available due to the data policy of PERSUADED project.

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
