# Peer review of "Toxicological Assessment of Oral Co-Exposure to Bisphenol A (BPA) and Bis(2-ethylhexyl) Phthalate (DEHP) in Juvenile Rats at Environmentally Relevant Dose Levels: Evaluation of the Synergic, Additive or Antagonistic Effects"

_ijerph, 2021, doi:10.3390/ijerph18094584_

Round 1
Reviewer 1 Report
The submitted paper is the logical consequence of having previously reported the same experimental approach but using BPA and DBP separately (references 10 and 11). For this reason, the discussion focuses on the comparison of the results obtained with respect to that reported by the same group previously. In my opinion, the most relevant part of the work is the use of computational tools applied to analyze combined effect of mixtures of chemicals commonly present in food and environment. However, the results obtained do not differ from those previously reported by other authors. As the author writes “adult male rats presented changes in lipid profile, liver related biochemical parameters, glucose and testosterone level, more pronounced in the mixture [reference 12]. Moreover, the BPA and DEHP mixture may disturb the thyroid hormone homeostasis in female rats treated during puberty [reference 13] ” in that sense there is no substantial contribution with respect to what has been reported on the subject.
Regardless, I consider that the work should be published because the results are part of a set of experiments that are carried out under the same conditions and therefore can be comparable.
Author Response
The Authors think that the manuscript represents a good example of the application of new approaches to the study of chemical mixtures.
They agree with the Reviewer that the results of the present paper will complete the evaluation of BPA, DEHP and their mixtures administered to rats during the juvenile period.
Reviewer 2 Report
Major revisions
- figure 1: left side figure: DEHP up to down should be narrow to wide?
- Ref. no. 15 is still in preparation: I do not know this is allowed to be included in the references. Need to check the journal policy concerning the citation of unpublished materials.
- Minor editorial revisions
- Line 68: Baralic et al. (2020) or Baralic et al. [13]
- Line 79: Tait et al. (2020, 2021) or Tait et al. [14.15]
- Lines 112, 143, 389: Tassinari et al. (2019, 2021) or Tassinari et al. [10, 11]
- Line 173: Estradiol should be E2
- Line 183: add city and country after Nikon Microphot FX
- Line 443: os?
- References should be carefully checked once again: titles should be small capitals, journal names should be appropriately abbreviated or full (need consistency, just follow the journal style), some references are missing volume and pages
- Line 615: bisphenol a should be bisphenol A
Author Response
Major revisions
- Figure 1: left side figure: DEHP up to down should be narrow to wide?
The Figure 1 has been modified accordingly
- Ref. no. 15 is still in preparation: I do not know this is allowed to be included in the references. Need to check the journal policy concerning the citation of unpublished materials.
The Authors asked the Editor. The journal publication policy allows the citation of the unpublished papers
Minor editorial revisions
- Line 68: Baralic et al. (2020) or Baralic et al. [13]
The citations were modified in the manuscript
- Line 79: Tait et al. (2020, 2021) or Tait et al. [14.15]
The citations were modified in the manuscript
- Lines 112, 143, 389: Tassinari et al. (2019, 2021) or Tassinari et al. [10, 11]
The citations were modified in the manuscript
All the following mistakes were amended in the manuscript; the References were carefully checked
- Line 173: Estradiol should be E2
- Line 183: add city and country after Nikon Microphot FX
- Line 443: os?
- References should be carefully checked once again: titles should be small capitals, journal names should be appropriately abbreviated or full (need consistency, just follow the journal style), some references are missing volume and pages
Line 615: bisphenol a should be bisphenol A
Reviewer 3 Report
This manuscript assesses the joint toxic effects of BPA and DEHP at environmentally relevant dose levels with oral co-exposure to juvenile rats. The study is interesting and can be accepted after some revisions.
- P1, line 24-25: The degree of synergy or antagonism were evaluated by synergy scores calculation, using present data and results of individually administered compounds. Please check the spelling of the sentence.
- Adding a graphical abstract to the article can make the research design ideas clearer.
- Please check the spelling of the sentence “Whole blood samples were allow to coagulate at room temperature for 1h, centrifuged twice for 15 min at 2000 rpm (Microlite Microfuge, Thermo Electron Corporation) 168 and stored at -80° C until use” in line 167-168, page 4.
- In the combined toxicology study, a single toxicant exposure group should be set in the group. Please explain the reasons for grouping in the study design and discuss them in the discussion.
- The author should explain why select mean ± standard error of the mean (SEM) instead of mean ± standard deviation (SD).
- The exposure duration for BPA in the manuscript is 28 days. Please describe the rationale for choosing this exposure duration.
- Please add the sequence of all primers in the experimental method.
- Please check the spelling of the sentence “this could be partially explained with the different mechanism at receptor level” in line 485, page 15.
- P15, line 501: Delete the redundant preposition “in”.
Author Response
This manuscript assesses the joint toxic effects of BPA and DEHP at environmentally relevant dose levels with oral co-exposure to juvenile rats. The study is interesting and can be accepted after some revisions.
- P1, line 24-25: The degree of synergy or antagonism were evaluated by synergy scores calculation, using present data and results of individually administered compounds. Please check the spelling of the sentence.
The sentence was modified
- Adding a graphical abstract to the article can make the research design ideas clearer.
The graphical abstract has been added to clarify the experimental design
- Please check the spelling of the sentence “Whole blood samples were allowto coagulate at room temperature for 1h, centrifuged twice for 15 min at 2000 rpm (Microlite Microfuge, Thermo Electron Corporation) 168 and stored at -80° C until use” in line 167-168, page 4.
The sentence has been modified
- In the combined toxicology study, a single toxicant exposure group should be set in the group. Please explain the reasons for grouping in the study design and discuss them in the discussion
The present study belongs to an experimental plan which includes the testing of three dose levels of BPA, three of DEHP and their mixtures (see Materials and Methods) for 28 days. The exposure to single and combined chemicals has been performed approximately at the same time (the experimental groups were shifted of several days to avoid problems at sacrifice) but the Authors decided to present and discuss the data individually due to the great number of endpoints and to better reveal the data. In addition, differently from most of the papers dealing with toxicological effects of chemical mixtures, more than one mixture has been tested in the experimental plan, that’s why they were chosen as topic of a dedicated paper to meet the complexity of the results. This has now been explained also in the manuscript.
- The author should explain why select mean ± standard error of the mean (SEM) instead of mean ± standard deviation (SD).
For graphical purposes both SEM and SD are widely accepted.
- The exposure duration for BPA in the manuscript is 28 days. Please describe the rationale for choosing this exposure duration.
The rats in the experiment with BPA, DEHP and their mixtures were exposed for 28 days from weaning to sexual maturity, according to the time schedule of the juvenile toxicity study (Narciso et al Reprod Toxicol. 2017 Sep;72:136-141.)
- Please add the sequence of all primers in the experimental method.
The sequences of all primers used in the study are described in the Table included in the Supplementary data
- Please check the spelling of the sentence “this could be partially explained with the different mechanism at receptor level” in line 485, page 15.
The sentence has been modified
- P15, line 501: Delete the redundant preposition “in”.
It was not redundant; the second ‘in’ belongs to the ‘in vivo’ term. Now it is in italics.